# The Influence of Noise in the Neurofeedback Training Sessions in Student Athletes

**DOI:** 10.3390/ijerph182413223

**Published:** 2021-12-15

**Authors:** Christophe Domingos, Higino da Silva Caldeira, Marco Miranda, Fernando Melício, Agostinho C. Rosa, José Gomes Pereira

**Affiliations:** 1Life Quality Research Centre, 2040-413 Rio Maior, Portugal; 2Laboratory of Physiology and Biochemistry of Exercise, Faculty of Human Kinetics, University of Lisbon, 1495-751 Oeiras, Portugal; higinocaldeira@campus.ul.pt (H.d.S.C.); jgpereira@fmh.ulisboa.pt (J.G.P.); 3Department of Physics, LaSEEB-System and Robotics Institute, Instituto Superior Técnico, University of Lisbon, 2695-066 Lisbon, Portugal; marco.miranda@tecnico.ulisboa.pt; 4Department of Bioengineering, LaSEEB-System and Robotics Institute, Instituto Superior Técnico, University of Lisbon, 2695-066 Lisbon, Portugal; fmelicio@deea.isel.ipl.pt (F.M.); acrosa@laseeb.org (A.C.R.); 5ISEL—Instituto Superior de Engenharia de Lisboa, Lisbon Polytechnic Institute, 1959-007 Lisbon, Portugal

**Keywords:** neurofeedback, noise, reaction time, working memory, athletes, performance

## Abstract

Considering that athletes constantly practice and compete in noisy environments, the aim was to investigate if performing neurofeedback training in these conditions would yield better results in performance than in silent ones. A total of forty-five student athletes aged from 18 to 35 years old and divided equally into three groups participated in the experiment (mean ± SD for age: 22.02 ± 3.05 years). The total neurofeedback session time for each subject was 300 min and were performed twice a week. The environment in which the neurofeedback sessions were conducted did not seem to have a significant impact on the training’s success in terms of alpha relative amplitude changes (0.04 ± 0.08 for silent room versus 0.07 ± 0.28 for noisy room, *p* = 0.740). However, the group exposed to intermittent noise appears to have favourable results in all performance assessments (*p* = 0.005 for working memory and *p* = 0.003 for reaction time). The results of the study suggested that performing neurofeedback training in an environment with intermittent noise can be interesting to athletes. Nevertheless, it is imperative to perform a replicated crossover design.

## 1. Introduction

Noise is a source of stress that is always present in the daily lives of the majority of the population and, depending on the presence or absence of noise, which has a direct impact on performance [1]. The interest in this subject mostly began with the discomfort that the Industrial Revolution caused in the workers [2] and questions began to emerge about the influence of noise and its impact on general health [3], general performance [4,5] and, most recently, sport performance [6]. Regarding sport performance, it is known that the noise does not seem to affect the components linked to the performance speed, but it decreases accuracy performance and short-term and/or working memory performance [7]. Additionally, the same authors outlined that people can develop more effective coping strategies when exposed to continuous noise of long-duration and even in intermittent noise of short-duration. Moreover, it is necessary to realize that the environment in which the athletes are performing is anticipates constant or intermittent noise in a multitude of ways. Whether in training or in competition, this noise variability may have a fundamental role in helping individuals explore their environment to collect information for actions [8]. One of the best examples that demonstrates that the absence of noise—which may not, however, be the most effective method for training athletes—has been the COVID-19 pandemic. Many athletes were confronted with the reality of empty stadiums and stands, and some solutions were implemented, such as the installation of immersive crowd noise sound systems in stadiums and arenas. The purpose of these systems was to provide athletes with a live, interactive sound ‘bubble’ that immerses players in realistic and responsive spectator reactions to the game. In this way, it is evident that noise affects human performance [9].

Neurofeedback training (NFT) has been an emerging technique applied in sport and has its main focus in self-regulation skills, such as reaction time and short-term memory who are essential for athletes’ performance [10,11]. Thus, the noise or the absence of the noise are factors that contributes positively or negatively to the performance and of the few studies done with NFT in athletes, none reports the influence of the noise in the performance of the athletes and the success of the NFT protocol that was applied [12,13].

The studies done with NFT in sports were performed without information about the athletes’ environmental conditions in which the and, knowing that noise is always present in their environment, it is imperative to understand whether neurofeedback training in the individual alpha band is a valid and efficient method for sports performance in an uncontrollable environment (i.e., a noisy room). Therefore, in order to clarify the impact of the noise in NFT sessions and sport performance, the aims of this study are to compare the impact that both intermittent noise and the absence of noise had on their individual alpha bands (IABs) in the NFT sessions and to understand if the athletes could improve performance with the intermittent noise protocol. To the best of our knowledge, no study has directly examined two similar protocols under different environmental conditions on sport. We hypothesize that (1) the silent room has the same results in the IAB compared with the noisy room (2) the silent room has the same results in performance tests compared with the noisy room. 

## 2. Materials and Methods

### 2.1. Subjects

A total of forty-five subjects (7 females) aged from 18 to 35 years old participated in the experiment (mean ± SD for age: 22.02 ± 3.05 years). Group 1 was composed of fifteen student athletes in noisy room (mean ± SD for age: 22.53 ± 3.89 years), group 2 was comprised of fifteen student athletes in silent room (mean ± SD for age: 22.33 ± 2.47 years) and control group was composed of fifteen student athletes (mean ± SD for age: 21.20 ± 2.62 years). No differences between group ages were found (*p* = 0.200). All student athletes have been involved in federated sports or practicing exercise or sport regularly for more than 5 years [14], contrary to the group of sedentary students who do not meet the minimum 5 times a week of at least moderate-intensity requirements to be considered active [15]. The inclusion criteria were as follows: (1) no history of psychiatric or neurological disorders; (2) no psychotropic medications or addiction drugs; (3) normal or corrected-to-normal vision; (4) minimum age of 18 years and maximum age of 35 years; and (5) practice moderate-intensity exercise at least 5 times a week (sport or gym) regardless of skill level (for athlete groups). Participants were randomized into three groups: (a) noisy room intervention group, (b) silent room intervention group, and (c) control group without sham. All students were informed about the possible risks of the investigation before providing written informed consent to participate. All procedures were approved by the Ethics Committee of the Faculty of Human Kinetics University of Lisbon and conducted in accordance with the Declaration of Helsinki [16]. Informed consent was obtained from all individual participants included in the study. All data collected has been stored in an encrypted database where only researchers related to the NFT project have access. Anonymity was guaranteed. 

### 2.2. Signal Acquisition

During the experiment, the participants sat in a room with a controlled environment. The EEG signals were recorded according to the international 10–20 system (Fp1, Fp2, F3, F4, F7, F8, C3, C4, T3, T4, P3, P4, T5, T6, O1, O2, Fz, Cz, and Pz), with a sampling frequency of 256 Hz, feedback was from Cz channel (it was chosen since it is at the primary motor cortex and has been associated with sensory information processing over the sensorimotor area and provide a measurement of the activity in both hemispheres and in the frontal lobe) [17,18], the ground was located at forehead and the reference was the average of left and right mastoids. The signals were amplified by a 24-channel system (Vertex 823 from Meditron Electomedicina Ltd.a, São Paulo, Brazil) and were recorded by Somnium software platform (Cognitron, São Paulo, Brazil) and NF module by Laseeb-ISR. Circuit impedance was kept below 10 kΩ for all electrodes before the sessions. Subjects were asked to sit comfortably and then to remain as still as possible and to avoid excessive blinking and abrupt movements.

### 2.3. Experimental Design

In the first session of this randomized controlled study, all intervention participants performed a 5-min NFT familiarization to understand how to achieve an alpha band mental state, followed by pre-testing (the performance tests are the same and will be described in the Assessments section). The pre- and post-tests had the same interval of time for both the control and intervention groups. Timeline of the NFT training sessions and respective performance tests (pre- and post-tests) are presented in Figure 1.

#### 2.3.1. Intervention Groups—Noisy Room and Silent Room

The intervention group performed a familiarization session and pre-tests before the 12 NFT sessions. At the end, post-tests were performed. The NFT sessions consisted in 25 trials of 60 s each with 5 s of pause between trials. The total NFT session time for each subject was 300 min. The NFT sessions were performed two times per week.

#### 2.3.2. Control Group

The control group only performed pre- and post-tests over a month and a half without the training sessions.

#### 2.3.3. Silent Room

Subjects were in a fully controlled environment. The room was isolated, no people around and no light. The subjects also used insulating headphones to muffle any possible noise.

#### 2.3.4. Noisy Room

The subjects and the investigator were in an unpredictable environment. The neurofeedback station was in a physiology laboratory. While the participants were doing the NFT, the researchers created three controlled noisy distractors situations to try to bring them closer to the sporting environment: (1) one of the researchers used the treadmill in a 12 km/h speed while speaking with the other research; (2) the researchers spook with each other near the participant and (3) one of the researchers use the treadmill without speech. We opted for intermittent noise rather than continuous noise due to the known detrimental effects on performance [7,19].

#### 2.3.5. Spectrogram

Ambient noise from both rooms were recorded for 5 min since there was a pattern during the 25 min of the sessions. After recording, the sound was transformed using SoX 2D monochrome spectrogram (Figure 2) [20]. The vertical axis of the spectrogram represents the frequency (kHz) and the horizontal axis represent the time. The shades of gray visualized is the noise recorded. Decibels relative to full scale (dBFS) is a unit of measurement for amplitude levels in digital systems, such as pulse-code modulation (PCM), which have a defined maximum peak level. The level of 0 dBFS is assigned to the maximum possible digital level.

### 2.4. Measurements

The baseline (IAB) was determined before and after NFT. The baseline recording consisted of four epochs of 30 s: two with eyes opened and two the eyes closed during the resting period. Recordings of eyes opened and closed in baseline 1 provide data for the calculation of alpha desynchronization and synchronization respectively, enabling to determine frequency bands individually through amplitude band crossings [11].

Feedback is a determinant step for the protocol’s success. Neural activity must be fed back by some parameter(s) and presented to the participant in a simple and direct representation of their value. In this study, the feedback parameter was the relative amplitude of the individual alpha band calculated as in Equation (1) where band amplitude was the amplitude of the IAB and EEG amplitude was the amplitude from 4 Hz to 30 Hz. Using the amplitude spectrum instead of the power spectrum prevents excessive skewing which results from squaring the amplitude, and thus increases statistical validity [21].
(1)$$Relative\;IAB\;amplitude=\frac{\frac{\sum_{k=LB}^{UB}X\left(k\right)}{UB\;-\;LB}}{\frac{\sum_{k=4}^{30}X\left(k\right)}{30\;-\;4}}$$

The visual feedback display contains two tridimensional objects: a sphere and a cube. The sphere radius reflects the feedback parameter value in real time and if it reaches a threshold (Goal 1) its colour changes. The sphere has several slices (initially four, the minimum), and the more present, the smoother it looks. While Goal 1 is being achieved slices are added, and if not, the sphere loses them until it has four again. The cube height is related to the period for which Goal 1 is achieved continuously. If it this exceeds a predefined period of time (2 s) Goal 2 is accomplished, and the cube rises until Goal 1 is failed. Then it starts falling until it reaches the bottom or Goal 2 is achieved again. Therefore, the participant’s task is to take the cube as high as possible [21].

The feedback threshold was set to 1.0 in the first session, and it will be adjusted according to the session report, which showed the percentage of time for which the feedback parameter was above the threshold in each session. If this percentage exceeded 60%, the threshold would be increased by 0.1 in the next session. In contrast, if the percentage was below 20%, the threshold would be decreased by 0.1 in the next session [22].

### 2.5. Assessments

#### 2.5.1. N-Back Test (Working Memory)

A sequence of random digits appears, with an interval of 2 s between each, and the subject must indicate (yes or no) whether the new number displayed is equal to any of the last 2 numbers presented immediately before. The sequence has 22 numbers [23].

#### 2.5.2. Oddball (Reaction Time)

The odd ball test is used to evaluate the attention of the subjects. In this test, different geometrical forms appear (circle, an octagon and a square) and the volunteers were instructed to click only if the circle appeared. The test consisted of 50 trials, where the images appeared for 0.5 s at an interval of 0.5 s. The defined decoy rate of 40% [24].

### 2.6. Statistical Analysis

Comparison of session SAB and IAB means between the two protocols was performed using the independent sample Student’s *t*-test and Mann–Whitney U test when normality was not verified. Differences in pre-test, post-test and the pre-to-post-test changes of SAB, IAB, NB and OB between the control group and the two protocols were examined using the ANOVA test; the Kruskal–Wallis test was performed when normality was not verified. To identify between which groups there were significant differences, a post hoc Tukey’s test was performed. Comparisons of group means between first and last session and pre-test and post-test (SAB, IAB, NB and OB) were analyzed using the paired sample Student’s *t*-test and Wilcoxon test when normality was not verified. A linear regression was performed to see the correlation between the variables (IAB and SAB) and time in the different groups. Sampling power was calculated using G*Power software (version 3.1.9.4) for a 0.05 significance level and a 0.95 power, resulting in a required sample of 45 participants. Data were analysed with SPSS software for Windows version 25.0 (SPSS Inc., Chicago, IL, USA). Statistical significance was set at *p* < 0.05 for all tests.

## 3. Results

Differences in SAB and IAB during NFT between the silent room and the noisy room groups are presented in Figure 3. In SAB, session 1 and session 12 showed significant differences between intervention groups (1.57 ± 0.08 vs. 1.08 ± 0.21, t(28) = −5.290; *p* < 0.001; 1.51 ± 0.09 vs. 1.15 ± 0.28, U = 45.000 *p* = 0.005, respectively). Similar results were verified for the IAB, in session 1 and session 12 between-intervention groups (1.56 ± 0.08 vs. 1.14 ± 0.26, t(28) = −4.039; *p* < 0.001; 1.59 ± 0.12 vs. 1.21 ± 0.31, U = 48.000 *p* < 0.001, respectively). For SAB, the noisy room protocol has a positive slope, while the silent room protocol has a negative slope. For IAB, the noisy room protocol has a slope much higher than the IAB for the silent room protocol (Z(1,10) = 4.812, *p* = 0.05; R^2^ = 0.33 vs. Z(1,10) = 0.219, *p* = 0.650; R^2^ = 0.02, respectively). Although no differences were found, only the noisy room presented a positive slope. 

Differences in performance tests between the silent room and the noisy room protocols and control group are presented in Table 1. Significant differences were found between control group and noisy room for the differences in NB (0.00 ± 1.83 vs. 9.33 ± 0; Z = −2.829; *p* = 0.005, respectively). The OB post-test showed significant differences more precisely between the control group and the silent room group (96.27 ± 0.78 vs. 98.40 ± 0.76; Z = −2.966; *p* = 0.008, respectively). 

Table 2 shows the differences between pre-tests and post-tests for each protocol and control group. Only in the noisy room protocol significant differences were found for the NB and OB between pre- and post-tests (*p* = 0.005 for NB; and *p* = 0.03 for OB).

## 4. Discussion

The first aim of the study was to clarify the importance of performing NFT sessions in a silent environment or in a noisy environment. The second aim of the study was to understand if athletes were able to improve on performance tests, even in noisy conditions.

The results of the study suggest that both the silent room and the noisy room had no results in increasing IAB and SAB after 12 sessions. The lack of improvements in alpha activity can be associated with the weekly training frequency. A recent study proved that a more frequent protocol (3 times/week) increased IAB, while performing only 2 sessions per week had no improvements in the same band [25]. This finding lends preliminary support to the null hypothesis that the SAB and IAB will be the same between both protocols, which means that the environmental condition where sessions were performed seems to not have an important contribution to the effectiveness of the training in relative alpha amplitude changes. On the other hand, the group that was in the noisy room appears to have significant results in all performance tests, which was not the case with the group that performed NFT sessions in the silent room. The null hypothesis was rejected by the differences in the performance tests.

Regarding the relative amplitude in both SAB and IAB, it can be verified that there seems to be an influence of the environment conditions on the initial value, i.e., the athletes who performed the training sessions in the silent room have initial values of relative amplitude higher than the group that held the sessions in the noisy room (1.57 ± 0.08 vs. 1.08 ± 0.21 for SAB; and 1.56 ± 0.08 vs. 1.14 ± 0.26 for IAB). These results may be associated to the environmental distractors, that is, more distractors (noisy room), could be associated to initial lower relative amplitude. Regarding the evolution of the relative amplitude over the sessions, it is possible to notice that only the group that was in the noisy room increased SAB and IAB and the group that was in the silent room only increased the IAB. These results reinforce the idea that it is necessary to use the IAB instead of the isolated SAB [11] and Figure 3 also allow us to realize that although the silent room increases the IAB, the slope is lower than that found by the group of the noisy room (R^2^ = 0.02 vs. R^2^ = 0.33, respectively). Even though the results were not significant in the athletes who were in the noisy room, it is possible to notice that they were able to, somehow, ignore the prejudicial noise during the sessions and focus on the concrete task.

All performance tests suggest that there were significant improvements in the group that did NFT sessions in a noisy environment. The NFT sessions seem to have been effective in improving scores on working memory and reaction time tests [26,27]. A possible cause for the controlled environment group not having significant results on the working memory and reaction time tests may be because they started with higher scores than the athletes in a noisy environment; that is, the margin of progression was lower. On the other hand, the post-tests scores of the noisy environment group were larger for the group in silent environment, which indicates that the distractor factor was aligned with the performance of the athlete due to their need to focus more on the task and inhibit the distractor.

The results obtained are in agreement with Jeon and colleagues (2014) and Balazova et al. (2008), who reported, in their studies, indications of the negative effects of distracting noise on sport performance; mental tasks seem to be effective in increasing performance since that helps the athlete to focus only on the main task [28,29]. On the other hand, several studies conducted in other populations indicate that noise has impaired working memory and reaction time [29]. It seems that the effect of room acoustics on human performance is dependent on the sound source and its relation to the job or task, on the task itself and on the personal factors of the subject performing the task [1].

The main strength of the study and what makes it so important is that it opens a door to the possibility of better understanding the question of replicability associated with the environment in which NFT sessions are performed; that is, in previous studies, the conditions of the environment wherein such sessions are held are not mentioned, and thus it is not known if the results obtained were influenced by the environment or not. The training individualization has also been considered (IAB was used instead of the SAB) but both IAB and SAB were present in the study [11]. A control group was used to ensure that learning depended on NFT and not on other factors. These last two arguments are two factors of robustness [12,13]. The point of this comparison between protocols is that it should emerge as a guideline for future investigation.

There are limitations that should be considered: (1) it is imperative to perform a replicated crossover design to be certain that the results are, in fact, influenced by the type of environment and not by other variables; (2) a questionnaire or scale is needed to better understand both what strategies athletes are using during NFT and mood [30] and (3) there is a large diversity of sports. The present study should therefore be considered exploratory.

## 5. Conclusions

The results obtained suggest that performing NFT sessions in a silent environment or in an environment with intermittent noise did not seem to have a direct impact on changes in the relative amplitude of SAB and IAB. However, the performance of the noisy-room group appeared to improve, given the results of their performance tests.

The study allows other researchers to realize that they can perform NFT sessions in both silent and/or noisy environment and, above all for athletes who are inserted in a noisy sporting context, NFT could help to improve reaction time and working memory.

Future research should replicate the noisy and silent rooms protocols based on a pre-test and post-test associated with the sport to better understand how the increased relative amplitude of the individual alpha band contributes to a better sporting performance. Likewise, it would be necessary to compare with a continuous noise environment. Finally, the results achieved need to be interpreted with caution since it was not a replicated crossover design.

## Figures and Tables

**Figure 1 ijerph-18-13223-f001:**
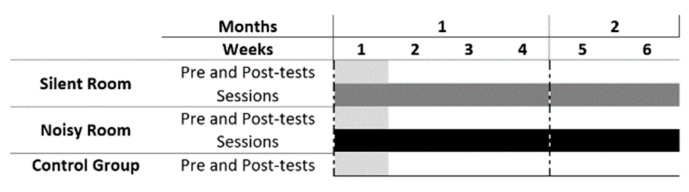
Timeline of the NFT training sessions and respective performance tests (pre and post tests).

**Figure 2 ijerph-18-13223-f002:**
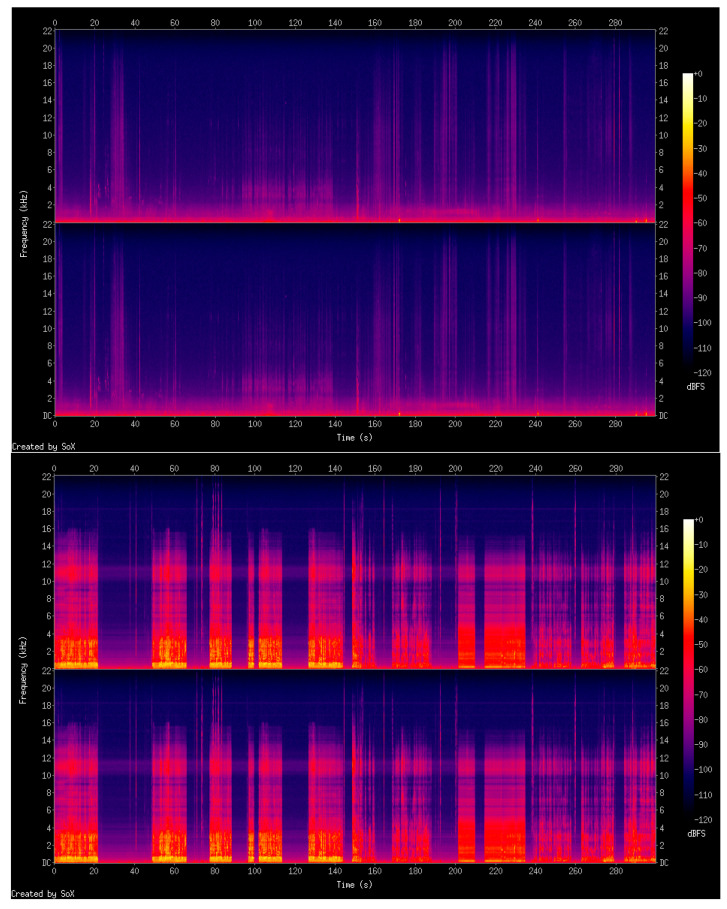
Spectrograms of the silent room (**upper figure**—peak level dB was −13.40) and the noisy room (**bottom figure**—peak level dB was −0.56).

**Figure 3 ijerph-18-13223-f003:**
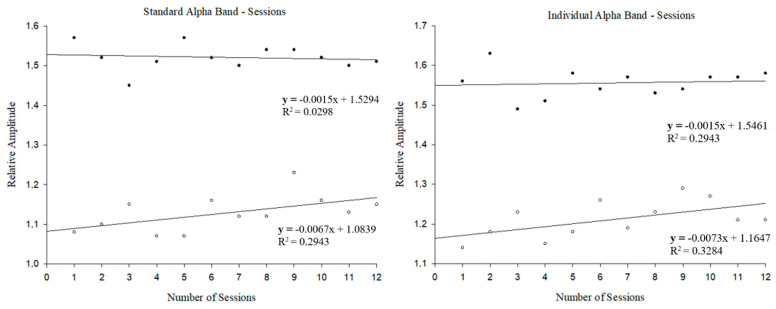
Differences between session 1 and 12 standard alpha band (**left image**) and individual standard alpha band (**right image**) for each protocol; filled dots represents the silent room group and empty dots represents the noisy room group.

**Table 1 ijerph-18-13223-t001:** Differences in standard alpha band (8 to 12 Hz), individual alpha band relative amplitude and performance tests between protocols.

		M ± SD		
	Control (*n* = 15)	Silent Room (*n* = 15)	Noisy Room (*n* = 15)	*p*
SAB S1	NA	1.57 ± 0.08	1.08 ± 0.21	<0.001 ^a^
SAB S12	NA	1.51 ± 0.09	1.15 ± 0.28	0.005 ^b^
Difference in SAB (S12–S1)	NA	−0.06 ± 0.06	0.07 ± 0.	0.113 ^a^
IAB session 1	NA	1.56 ± 0.08	1.14 ± 0.26	<0.001 ^a^
IAB session 12	NA	1.59 ± 0.12	1.21 ± 0.31	0.007 ^b^
Difference in IAB (S12–S1)	NA	0.04 ± 0.08	0.07 ± 0.	0.740 ^b^
NB pre-test	96.00 ± 3.87	93.67 ± 6.94	89.33 ± 9.23	0.127 ^d^
NB post-test	96.00 ± 6.32	96.33 ± 3.99	98.67 ± 8.84	0.189 ^d^
Difference in NB (post-test–pre-test)	0.00 ± 0.07	2.67 ± 6.78	9.33 ± 8.84	0.005 ^c^
OB pre-test	94.80 ± 5.28	96.27 ± 3.20	95.20 ± 3.84	0.724 ^d^
OB post-test	96.27 ± 3.01	98.40 ± 2.95	98.53 ± 1.41	0.008 ^d^
Difference in OB (post-test–pre-test)	1.47 ± 2.77	2.13 ± 3.58	3..33 ± 3.44	0.298 ^c^

M, mean; SD, standard deviation; SAB, standard alpha band; IAB, individual alpha band; S1, session 1; S12, ses-sion12; NB, n-back test; OB, oddball test; NA, Not Applicable. ^a^ Differences between groups tested with Student’s *t*-test. ^b^ Differences between groups tested with Mann–Whitney U. ^c^ Differences between groups tested with ANOVA. ^d^ Differences between groups tested with Kruskal–Wallis Test.

**Table 2 ijerph-18-13223-t002:** Differences between pre-tests and post-tests for each protocol.

	M ± SD		
	Pre-Test	Post-Test	*p*
Control			
NB	96.00 ± 3.87	96.00 ± 6.32	0.666 ^a^
OB	94.80 ± 5.28	96.27 ± 3.01	0.059 ^a^
Silent Room protocol			
NB	93.67 ± 6.93	96.33 ± 3.99	0.142 ^a^
OB	96.26 ± 3.20	98.40 ± 2.95	0.057 ^a^
Noisy Room protocol			
NB	89.33 ± 9.23	98.67 ± 2.97	0.005 ^a^
OB	95.20 ± 3.84	98.53 ± 1.41	0.003 ^a^

M, mean; SD, standard deviation; NB, n-back test; OB, oddball test; ^a^ Differences tested with Wilcoxon test.

## Data Availability

The data presented in this study are available on request from the corresponding author.

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
