# Peer review of "The Influence of Noise in the Neurofeedback Training Sessions in Student Athletes"

_ijerph, 2021, doi:10.3390/ijerph182413223_

Round 1
Reviewer 1 Report
Neurofeedback training (NFT) enables users to learn self-regulation of their cortical oscillations by receiving moment-to-moment feedback from their electroencephalogram (EEG). NFT is a safe, inexpensive, and accessible technology that is a valuable intervention. Several lines of evidence have demonstrated NFT as a promising and nonpharmacological supportive treatment for neurological and psychiatric disorders, such as attention deficit hyperactivity disorder (ADHD), depression, anxiety, or insomnia. NFT has also been applied in healthy participants to enhance several aspects of cognitive functions. In my opinion, the results obtained do not bring anything innovative. Neurofeedback training is known to be more effective when done in silence. Patients should stayed in a quiet room and sat comfortably with apparatus. The training is very efficient and effective when the patient is able to concentrate and focus. The results obtained completely distort this concept and I do not fully agree with them. In my opinion, you should chande the aim of work.
Author Response
Reviewer_1:
Point 1.
1.1. Neurofeedback training (NFT) enables users to learn self-regulation of their cortical oscillations by receiving moment-to-moment feedback from their electroencephalogram (EEG). NFT is a safe, inexpensive, and accessible technology that is a valuable intervention. Several lines of evidence have demonstrated NFT as a promising and nonpharmacological supportive treatment for neurological and psychiatric disorders, such as attention deficit hyperactivity disorder (ADHD), depression, anxiety, or insomnia. NFT has also been applied in healthy participants to enhance several aspects of cognitive functions. 1.2. In my opinion, the results obtained do not bring anything innovative. Neurofeedback training is known to be more effective when done in silence. Patients should stayed in a quiet room and sat comfortably with apparatus. The training is very efficient and effective when the patient is able to concentrate and focus. 1.3. The results obtained completely distort this concept and I do not fully agree with them. In my opinion, you should chande the aim of work.
Response 1.
1.1. We want to thank you for your suggestions and sincere opinion about our work.
1.2. We respect your honest opinion, but our results shown exactly why this is an innovative work. Nevertheless, we do agree that we lack a lot of background in the introduction to support our results, so we assume full responsibility of it and will improve our introduction to help understanding our outcomes:
“Additionally, the same authors, outlined that people can develop more effective coping strategies when exposed to continuous noise of long-duration and even in intermittent noise of short-duration Moreover, it is necessary to realize that the environment in which the athletes are performing is slated to constant or intermittent noise in all the possible ways. Whether in their training or in their competitions, this noise variability may have a fundamental role in helping individuals explore their environment to collect information for actions [8]. One of the best examples that demonstrates that the absence of noise may not be the most effective method for training athletes is the COVID-19 pandemic. Many athletes were confronted with the reality of empty stadiums and stands, and some solu-tions were implemented, such as the installation of immersive crowd noise sound sys-tems in stadiums and arenas. The purpose of these systems was to provide athletes with a live, interactive sound 'bubble' that immerses players in realistic and responsive spectator reactions to the game. In this way, it is evident that noise affects human performance [9].”
Furthermore, we will add to the limitation the need of a group in a continuous noise environment.
“On the one hand, the absence of noise seems to have a positive effect on performance [13] and on the other hand, the noise seems to have a negative impact in performance [14, 15].” This paragraph was excluded because it is redundant (with the information added previously) and can cause confusion with the aim of the study.
1.3. Again, with the background added to the introduction we hope that our objectives are clearer. Athletes cannot be compared to clinical or non-athlete populations as it was seen in this recent study comparing athletes with non-athletes during an alpha band neurofeedback training [1].
- Domingos, C., C.P. Alves, E. Sousa, A. Rosa, and J.G. Pereira, Does Neurofeedback Training Improve Performance in Athletes? NeuroRegulation, 2020. 7(1): p. 8-8.

Reviewer 2 Report
Trainability was highly personalized, especially in neurofeedback training, replicate cross design and longitudinal research was more suitable for neurofeedback training.
There was insufficient background about the literature of training in noise environment. Training in long time noise environment with negative performance was mentioned, however there was no literature to support the short period noise will impact the training performance.
The variables that short period noise impacted the training performance was unclear, normal training without neurofeedback training needed to support that the short period noise was the main impact factor of the negative performance.
Whether it was a randomized controlled trial was unclear.
The total time and the decibel of the intermittent noise were not clear, which was the main impact factors for the study.
The details of the study experiment describing were not clear, for example, 2.3.4 Athletes were tested on treadmill in laboratory in noisy environment. How to test on treadmill was not clearly described.
Serial number error and some repeated title were detected in the study.
Author Response
Reviewer_2
Point 1.
Trainability was highly personalized, especially in neurofeedback training, replicate cross design and longitudinal research was more suitable for neurofeedback training.
Response 1.
We appreciate your time and suggestions, as they have provided valuable help towards improving our work.
Point 2.
There was insufficient background about the literature of training in noise environment. Training in long time noise environment with negative performance was mentioned, however there was no literature to support the short period noise will impact the training performance.
Response 2.
We do agree that we lack a lot of background in the introduction to support our results, so we assume full responsibility of it and will improve our introduction to help understanding our outcomes:
“Additionally, the same authors, outlined that people can develop more effective coping strategies when exposed to continuous noise of long-duration and even in intermittent noise of short-duration Moreover, it is necessary to realize that the environment in which the athletes are performing is slated to constant or intermittent noise in all the possible ways. Whether in their training or in their competitions, this noise variability may have a fundamental role in helping individuals explore their environment to collect information for actions [8]. One of the best examples that demonstrates that the absence of noise may not be the most effective method for training athletes is the COVID-19 pandemic. Many athletes were confronted with the reality of empty stadiums and stands, and some solutions were implemented, such as the installation of immersive crowd noise sound systems in stadiums and arenas. The purpose of these systems was to provide athletes with a live, interactive sound 'bubble' that immerses players in realistic and responsive spectator reactions to the game. In this way, it is evident that noise affects human performance [9].”
Furthermore, we will add to the limitation the need of a group in a continuous noise environment.
“On the one hand, the absence of noise seems to have a positive effect on performance [13] and on the other hand, the noise seems to have a negative impact in performance [14, 15].” This paragraph was excluded because it is redundant (with the information added previously) and can cause confusion with the aim of the study.
Point 3.
The variables that short period noise impacted the training performance was unclear, normal training without neurofeedback training needed to support that the short period noise was the main impact factor of the negative performance.
Response 3.
As mentioned above, we hope that the information added to the introduction will make the objectives of the study clearer. The introduction was confusing and even contradictory to our goals.
Point 4.
Whether it was a randomized controlled trial was unclear.
Response 4.
In the subsection “2.1. – Subjects” you can find the information:
“Participants were randomized into three groups: a) noisy room intervention group, b) silent room intervention group, and c) control group without sham.”
Point 5.
The total time and the decibel of the intermittent noise were not clear, which was the main impact factors for the study.
Response 5.
We agree with you and since this is an open journal, we will replace Figure 2. for the same with colors. Moreover, we will include some written information regarding the maximal values attained: “Figure 2. Spectrograms of the silent room (upper figure – peak level dB was -13.40) and the noisy room (bottom figure – peak level dB was -0.56).”
Point 6.
The details of the study experiment describing were not clear, for example, 2.3.4 Athletes were tested on treadmill in laboratory in noisy environment. How to test on treadmill was not clearly described.
Response 6.
Thank you again. We will include more information to be more perceptible for readers:
While the participants were doing the NFT, the researchers created 3 controlled noisy dis-tractors situations to try to bring them closer to the sporting environment: 1) one of the re-searchers used the treadmill in a 12km/h speed while speaking with the other research; 2) the researchers spook with each other near the participant and 3) one of the researchers use the treadmill without speech.
Point 7.
Serial number error and some repeated title were detected in the study.
Response 7.
We would be immensely grateful if you could specifically mention the errors and repeated tittles. Maybe in the original paper?

Reviewer 3 Report
Abstract
- The age range is too wide not to have changes due to the age of the participants.
- Were they equally divided by age in a homogeneous way or only by number?, i.e., if out of 45 students it was divided into three bearing in mind the distribution by age or not.
- The intervention is 300 minutes long? Isn’t it too long?
- The intervention was done twice a week, did you follow any previous study? Why two days a week?
- Studies regarding the fact that the environmental condition does not have an important contribution on the effectiveness of the treatment. Results contradict this statement. In the results, why does noise improve the results? How was isolation achieved or how was it done in order not to have that noise? Which studies are there about this?
Introduction
- Results are opposite to the ones presented in your result section. Why? How do you account for this fact?
- References to 13 and 14 are once again results contrary to the ones obtained in this research.
- Hypothesis 2 – why differences are not presupposed here?
Materials and methods
- 45 subjects are not much for an RTC, how was the sample size calculated? Why 45? How was the sample chosen? Techniques…
- There should be the same proportion of men and women. How can it be concluded then if there are gender differences if the sample is not representative?
- Are there not differences based on the sport dealt with? A homogeneous sample should be taken devoted to similar activities.
- When you state that randomization was done randomly in three groups… In which way was it done? How did you carry out the randomization?
- At the end of the paragraph you state that anonymity was guaranteed. However, how was it done?
- Patients were asked to sit comfortably. During the intervention, what were patients supposed to do? Which orders did they receive?
- All the participants, including the control group, took the familiarization?
- Did you follow previous recommendations from any other research?
- As for the duration of the intervention… 5 hours of intervention? How did you do for participants not to feel tired? Did you schedule breaks during the measurement? How do you know that such a long period of intervention is not having any effect? In fatigue and other related parameters.
- Why was it done without light?
- In reference to the noisy room – intermittent noise could be beneficial but the constant one could not? How can you tell the differences between each other?
- For the measurements, at the end of the sentence “Using the amplitude spectrum instead of the power spectrum prevents excessive skewing which results from squaring the amplitude, and thus increases statistical validity.”. References are missing.
- Within the statistical analysis – the calculation of the sample size should also be included before. A regression analysis could have been added? How was the randomization of the groups blinded? Did researchers know in which one each one was? Data were analyzed by the same people who collected them? All of this should be explained.
Results:
- There should be further explanation in this section. It is very short compared to the rest.
Discussion
- Only by augmenting one it will significantly improve? Which is the explanation behind?
- Hypothesis are not usually posed as they will have the same result. It seems that in this case results were obtained first and later on the hypothesis was posed, taking into consideration that there are no differences.
- As for the SAB and IAB, why does it improve in one parameter and not in the other one?
- How did participants ignore the noise? How can you account for this?
- “A possible cause for the controlled environment group do not have significant results in the working memory and reaction time tests may be because they started with higher scores than the athletes in a noisy environment, that is, the margin of progression was lower.”. This makes them not comparable.
- “The distractor factor ends up allied to the perfor-mance of the athlete because they need to focus more on the task and inhibit the distractor.” Is there any theory or reference explaining this?
- In Balazova et. al (2008) quote, the period (.) is written after ‘al.’
- In reference 30, which methodological differences are between each other?
- It seems that the effect of room acoustics on human per-formance is dependent on the sound source and its relation to the job or task, on the task itself and on the personal factors of the subject performing the task [1]. This provokes that everything should be controlled in the study.
- I agree with your limitations. Totally, sport people with such different conditions cannot be compared. Besides, the sample size is the one for a pilot study.
- There are many factors that could have influenced so that results could considered non-conclusive.
Author Response
Reviewer_3:
Point 1.
Abstract
1.1 The age range is too wide not to have changes due to the age of the participants.
1.2 Were they equally divided by age in a homogeneous way or only by number?, i.e., if out of 45 students it was divided into three bearing in mind the distribution by age or not.
1.3 The intervention is 300 minutes long? Isn’t it too long?
1.4 The intervention was done twice a week, did you follow any previous study? Why two days a week?
1.5 Studies regarding the fact that the environmental condition does not have an important contribution on the effectiveness of the treatment. Results contradict this statement. In the results, why does noise improve the results? How was isolation achieved or how was it done in order not to have that noise? Which studies are there about this?
Response 1.
We want to thank you for your suggestions and questions. It's really rewarding to see that the issues raised will help to greatly improve our work!
1.1 We agree that we didn’t have the significance between ages in our methodology, however, it seem superficial to consider promptly that because of the wide range there are automatically significance between subjects. For that reason, we added the significance in the subject’s subsection:
1.2 They were distributed randomly. We only had one subject with more than 30 years. As you can find in the Subjects subsection, the mean age ranges between 21 and 22 years old.
1.3 The 300 min are the total time. You can find the information detailed in the Intervention group: “The intervention group performed a familiarization session and pre-tests before the 12 NFT sessions. At the end, post-tests were performed. The NFT sessions consisted in 25 trials of 60 s each with 5 s of pause between trials. The total NFT session time for each subject was 300 min. The NFT sessions were performed 2 times per week”.
1.4 This study was made accordingly to a wining project where we tested several methodologies in athletes using neurofeedback. The total sample of the project was 90, but for testing the noise vs silence we selected 2 weekly sessions since is the most reported in the literature (athletes doing NFT).
1.5 We indeed lacked a solid background in our introduction, but crucial changes were made to clarify our aims and results.
Point 2.
Introduction
1.1 Results are opposite to the ones presented in your result section. Why? How do you account for this fact?
1.2 References to 13 and 14 are once again results contrary to the ones obtained in this research.
1.3 Hypothesis 2 – why differences are not presupposed here?
Response 2.
2.1. As mentioned previously, we improved our introduction to clarify our results.
“Additionally, the same authors, outlined that people can develop more effective coping strategies when exposed to continuous noise of long-duration and even in intermittent noise of short-duration Moreover, it is necessary to realize that the environment in which the athletes are performing is slated to constant or intermittent noise in all the possible ways. Whether in their training or in their competitions, this noise variability may have a fundamental role in helping individuals explore their environment to collect information for actions [8]. One of the best examples that demonstrates that the absence of noise may not be the most effective method for training athletes is the COVID-19 pandemic. Many athletes were confronted with the reality of empty stadiums and stands, and some solu-tions were implemented, such as the installation of immersive crowd noise sound sys-tems in stadiums and arenas. The purpose of these systems was to provide athletes with a live, interactive sound 'bubble' that immerses players in realistic and responsive spectator reactions to the game. In this way, it is evident that noise affects human performance [9].”
Furthermore, we will add to the limitation the need of a group in a continuous noise environment.
“On the one hand, the absence of noise seems to have a positive effect on performance [13] and on the other hand, the noise seems to have a negative impact in performance [14, 15].” This paragraph was excluded because it is redundant (with the information added previously) and can cause confusion with the aim of the study.
2.2. Answered above.
2.3. We present the hypothesis in a statistical way, that is, the hypothesis written belongs to the null hypothesis.
Point 3:
Materials and methods
3.1 45 subjects are not much for an RTC, how was the sample size calculated? Why 45? How was the sample chosen? Techniques…
3.2 There should be the same proportion of men and women. How can it be concluded then if there are gender differences if the sample is not representative?
3.3 Are there not differences based on the sport dealt with? A homogeneous sample should be taken devoted to similar activities.
3.4 When you state that randomization was done randomly in three groups… In which way was it done? How did you carry out the randomization?
3.5 At the end of the paragraph you state that anonymity was guaranteed. However, how was it done?
3.6 Patients were asked to sit comfortably. During the intervention, what were patients supposed to do? Which orders did they receive?
3.7 All the participants, including the control group, took the familiarization?
3.8 Did you follow previous recommendations from any other research?
3.9 As for the duration of the intervention… 5 hours of intervention? How did you do for participants not to feel tired? Did you schedule breaks during the measurement? How do you know that such a long period of intervention is not having any effect? In fatigue and other related parameters.
3.10 Why was it done without light?
3.11 In reference to the noisy room – intermittent noise could be beneficial but the constant one could not? How can you tell the differences between each other?
3.12 For the measurements, at the end of the sentence “Using the amplitude spectrum instead of the power spectrum prevents excessive skewing which results from squaring the amplitude, and thus increases statistical validity.”. References are missing.
3.13 Within the statistical analysis – the calculation of the sample size should also be included before. A regression analysis could have been added? How was the randomization of the groups blinded? Did researchers know in which one each one was? Data were analyzed by the same people who collected them? All of this should be explained.
Response 3.
3.1. We understand your point, however we are not considering a normal or clinical population. We are considering athletes. We performed sampling power before the intervention (you can find this info in the Statistical Analysis subsection) for a 0.05 significance level and a 0.95 power. Additionally, I would like to mention that our study, with 45 people, is one of the biggest considering athletes and neurofeedback training. At least until late 2019.
3.2. We agree with your statement. It would be necessary to perform an analysis for the gender interaction. Since the other 2 reviewers didn’t mention results modifications, I would not present the statistical analysis in this review round.
3.3. Indeed, it would be better a homogeny in our sample, but as we explained in our methodology, we used the Cz channel to avoid that question. You can find full explanation in signal acquisition subsection.
3.4. We used the 6-steps randomization: 1) defining the population; 2) calculating our sample size; 3) listing the population; 4) assigning numbers; 5) finding random numbers; and 6) selecting from that.
3.5. It was stored in an encrypted folder where only the main author has access (me).
3.6. To sit comfortably and to do the NFT the way it is explained in the measurement’s subsection.
3.7. As is written in the control group subsection, they only did the pre and post-tests, so it was not necessary to do familiarization with NFT because they did not perform a single session.
3.8. The NFT in sports is scarce. We based our information in Mirifar and Xiang revisions.
3.9. 5h duration total. Not in a single session. Each session was 25min. The information is explained in the experimental design.
3.10. Light is a distractor and even more when we are testing alpha band. Alpha band is well known because of their ERD/ESD potentials.
3.11. This is a good question and pertinent. We only can infer that conclusion with another group performing constant noise. That’s why we added that information to our limitations.
3.12. Thank you. The references were placed.
3.13. The power sample was made and referred in the paper (see statistical subsection, please). We forgot to mention that we used the correlation, thank you for noticing (information was add). We never wrote that it was a blind randomization (if one group was placed in a silent room and the other in a noisy room, it would be obvious for participants and researchers – once again, that’s why we wrote in the limitation the need of a sham control group).
Point 4.
Results:
There should be further explanation in this section. It is very short compared to the rest.
Response 4.
We will add more statistical information to the ones already presented.
Point 5.
Discussion
5.1 Only by augmenting one it will significantly improve? Which is the explanation behind?
5.2 Hypothesis are not usually posed as they will have the same result. It seems that in this case results were obtained first and later on the hypothesis was posed, taking into consideration that there are no differences.
5.3 As for the SAB and IAB, why does it improve in one parameter and not in the other one?
5.4 How did participants ignore the noise? How can you account for this?
5.5 “A possible cause for the controlled environment group do not have significant results in the working memory and reaction time tests may be because they started with higher scores than the athletes in a noisy environment, that is, the margin of progression was lower.”. This makes them not comparable.
5.6 “The distractor factor ends up allied to the perfor-mance of the athlete because they need to focus more on the task and inhibit the distractor.” Is there any theory or reference explaining this?
5.7 In Balazova et. al (2008) quote, the period (.) is written after ‘al.’
5.8 In reference 30, which methodological differences are between each other?
It seems that the effect of room acoustics on human per-formance is dependent on the sound source and its relation to the job or task, on the task itself and on the personal factors of the subject performing the task [1]. This provokes that everything should be controlled in the study.
5.9 I agree with your limitations. Totally, sport people with such different conditions cannot be compared. Besides, the sample size is the one for a pilot study.
5.10 There are many factors that could have influenced so that results could considered non-conclusive.
Response 5.
5.1. We are sorry, but when you say “augmenting one” are you referring at which variable? Only performance tests had significant results, so I will assume that you are talking about NB and OB variables. In this case, the prove is the statistic who shows significant differences.
5.2. I already mentioned this topic above. We wrote this paper considering our hypothesis as the statistic is. H0 and H1:
H0: the silent room has the same results in the IAB compared to the noisy room
H1: the silent room has different results in the IAB compared to the noisy room
----
H0: the silent room has the same results in performance tests compared to the noisy room.
H1: the silent room has different results in performance tests compared to the noisy room.
We always want to reject the H0.
5.3 SAB is the Standard alpha band (8 to 12 Hz) and IAB is the individual alpha band. We demonstrate the importance of individualizing the NFT because most of the literature is based in the SAB. And as you can see, it is not the right choice. We (researchers using NFT) need to be cautious and try to individualize the training or we will present results with bias. This is a warning to other peers-
5.4. At the end of each session, we asked what the strategy was used. Only in the first and second sessions (and not all) noticed the noise. Athletes can easily adapt. That’s why we can’t compare athletes with non-athletes or clinical populations [1].
5.5 I will rectify the sentence because I’m saying, “compared with the noisy group”, but I never compare groups in time. The evolution from session 1 to session 12 is within group and not between.
5.6. We already added it in the introduction with the new paragraph.
5.7. Thank you, we will correct.
5.8. We are sorry, but are you referring only to the methodology of Balazova paper or are you asking between her methodology and mine or between her methodology and Jeon? We get your point, and we add information in methodology who explains how we tried to control the environment, too. Nevertheless, I have to say that athletes are performing (training and competition) in uncontrolled environment. The ecology of environment is essential.
5.9. Again, the power sample was made, and we are one of the studies with the biggest number of participants (until 2019, only 2 or 3 studies had more than us). Nevertheless, we can change our title and refer that is a pilot study.
5.10. In all your statistical commentaries, we agree that it could be wise to see the interaction between gender and our outcomes (even knowing a priori it will be hard that 7 females would contribute to alter significantly the results). Regarding the other statistical outcomes, we tried to explain the best we could.
- Domingos, C., C.P. Alves, E. Sousa, A. Rosa, and J.G. Pereira, Does Neurofeedback Training Improve Performance in Athletes? NeuroRegulation, 2020. 7(1): p. 8-8.

Round 2
Reviewer 1 Report
Dear All Authors,
now, your manuscript looks better. I recommend it for publication in this version.
Good luck.
Author Response
Dear reviewer,
We want to thank you the time spent reviewing and helping improve our work.
Our best regards.
Reviewer 2 Report
Thank you for your quick reply and work.
I am sorry that the revised paper can not cover the expectations.
The design of the study was deficient,replicate cross design and longitudinal research was more suitable for neurofeedback training.
If you insist on this study design, maybe you should change the aim of work.
Author Response
Dear reviewer,
We want to thank you for all your feedback. We'll try to answer to your concerns.
The study design that you indicate is really the most complete and the one that really fit the best. Unfortunately, we've already applied the study with this methodological design to 45 people. We only have 2 solutions:
1) redo this entire study with the methodology you presented;
2) accept your suggestions and be more cautious with the results we got.
I've no doubts that we'll always consider your design suggestion in future investigations, however, in this case we'll proceed with option 2.
Abstract (changes):
We tried to soften our aim.
"Considering that athletes constantly practice and compete in noisy environments, the aim was to investigate if performing neurofeedback training in these conditions would yield better results in performance than in silent ones."
We started to use more suggestive (cautious) words to explain our results: "appears", "seem", "suggested", "can be".
"The environment in which the neurofeedback sessions were conducted did not appear to have a significant impact on the training's success in terms of alpha relative amplitude changes (0.04 ± 0.08 for silent room versus 0.07 ± 0.28 for noisy room, p=0.740). However, the group exposed to intermittent noise appears to have favorable results in all the performance assessments (p = 0.005 for working memory and p = 0.003 for reaction time). "
"Nevertheless, it is imperative to perform a replicated crossover design. ".
Discussion:
We changed the words X -» Y (as follow):
- demonstrate -» suggest;
- has -» seems to not have;
- had -» appears to have;
We rewrote the main strength:
"The main strength of the study and what makes it so important is that it opens a door to the possibility of better understanding the question of replicability associated to the environment where the NFT sessions are performed, "
We added information to the limitations:
We mentioned the importance of re-doing a research with a replicated crossover design:
"There are limitations that should be considered: 1) it is imperative to perform a replicated crossover design to be certain that the results are in fact influenced by the type of environment and not by other variables; "
Conclusions:
We did fundamental changes to aware readers that the results must be interpreted with caution:
"The results obtained suggest that performing NFT sessions in a silent environment or in an environment with intermittent noise did not seem to have direct impact on changes in the relative amplitude of SAB and IAB. However, the group that was in the noisy room appeared to improve the results in the performance tests.
The study allows other researchers to realize that they can perform NFT sessions in both silent and/or noisy environment and above all for athletes who are inserted in a noisy sporting context, NFT could help to improve reaction time and working memory.
Future research should replicate the noisy and silent rooms protocols based on a pre-test and post-test associated to the sport to better understand how the increased relative amplitude of the individual alpha band contributes to a better sporting performance. Likewise, it would be necessary to compare with a continuous noise environment. Finally, the results achieved need to be interpreted with caution since it was not a replicated crossover design."
Reviewer 3 Report
Dear authors,
Thank you for considering the recommendations. We hope they have served to rethink certain aspects and they are useful for future studies.
All the best
Author Response
Dear reviewer,
We want to thank you for all your suggestions. They improved our work.
Best regards.